# ATTENDING: FEDERATED LEARNING WITH PERSONALIZED ATTENTIVE PRUNING FOR HETEROGENEOUS CLIENTS

## ABSTRACT

Federated Learning (FL) emerges as a novel machine learning paradigm, enabling distributed clients to collaboratively train a global model while eliminating local data transmission. Despite its advantages, FL faces challenges posed by system and data heterogeneity. System heterogeneity prevents low-end clients from participating in FL with uniform models, while data heterogeneity adversely impacts the learning performance of FL. In this paper, we propose the personalized ATTENtive pruning enabled federateD learnING (ATTENDING) to collectively address these heterogeneity challenges. Specifically, we first design an attention module incorporating spatial and channel attention to enhance the learning performance on heterogeneous data. Subsequently, we introduce the attentive pruning algorithm to generate personalized local models guided by attention scores, aiming to facilitate clients' participation in FL. Finally, we introduce a specific heterogeneous aggregation algorithm integrated with an attention matching mechanism to efficiently aggregate the pruned models. We implement ATTENDING with a real FL platform and the evaluation results show that ATTENDING significantly outperforms the baselines by up to 11.3% and reduces the average model footprints by 32%. Our code is available at: `https://anonymous.4open.science/r/ATTENDING`.

## 1 INTRODUCTION

The proliferation of smart devices and intelligent applications has significantly increased the volume of data generated at the edge of networks. In response to this trend, Federated Learning (FL) has emerged as a promising paradigm for collaboratively learning from geographically distributed data (Khan et al., 2021; Soltani et al., 2023; Liu et al., 2024; Wang et al., 2024a). Compared to traditional centralized machine learning approaches, FL effectively mitigates the systemic privacy risks and avoids prohibitively high costs associated with transmitting raw data. Owing to these advantages, FL has garnered significant attention and has been successfully implemented in numerous intelligent applications (Khan et al., 2021; Soltani et al., 2023).

Despite its advantages, FL still faces two critical challenges: **System heterogeneity** and **Data heterogeneity** (Lim et al., 2020; Khan et al., 2021; Liu et al., 2022; Chen et al., 2023; Wang et al., 2023a;b; 2024b; Li et al., 2024a). **System heterogeneity** refers to variations in device capabilities (e.g., CPU state, memory capacity, battery level, etc.) (Alam et al., 2022; Jiang et al., 2022b; Li et al., 2024b). In classical FL, the devices (i.e., clients) are required to update local models that share an identical footprint as the global model, leading to the failure of local model updates at clients with weaker capabilities (i.e., low-end clients). **Data heterogeneity** refers to different distributions and/or amounts of the local data on various clients, i.e., non-Independent and Identically Distributed (non-IID) data (Zhao et al., 2018; Li et al., 2022; Ma et al., 2024). Clients possessing only a small sample of data with uncommon labels will gain no benefit from FL since the data on other clients contain different distributions. To enhance clarity, we have elaborated on these two challenges by providing an example FL process in the context of the Internet of Things (IoT) network in Appendix A.

To mitigate the aforementioned challenges, several pioneering work propose to optimize local updating procedures to allow low-end clients to participate in FL training (Li et al., 2020; Karimireddy

et al., 2020; Wang et al., 2020b). Nonetheless, these optimization-based approaches focus on minimizing computational resource consumption, without considering the conservation of storage resources in mobile devices (Sun & Wei, 2022; Yang et al., 2022; Zhu et al., 2023). Therefore, model pruning techniques have recently been employed in FL for shrinking the footprint of models, consequently reducing the consumption of both computational and storage resources (Horvath et al., 2021; Wu et al., 2021; Li et al., 2021; Jiang et al., 2022a;b; Yi et al., 2024). Model pruning-based approaches typically prune out unnecessary local model parameters on individual clients, while maintaining a binary mask matrix for each client to indicate the presence of the corresponding parameters. However, existing model pruning-based FL approaches typically aggregate pruned models through element-wise averaging with the mask matrices. Unfortunately, as demonstrated in (Wang et al., 2020a; Jiang et al., 2022a), this strategy often yields detrimental effects on the model accuracy owing to the presence of permutation invariance in neural networks. Moreover, the computation cost of this strategy increases rather than decreases (Jiang et al., 2022a). This phenomenon arises from the fact that, within prevailing deep learning frameworks (e.g., PyTorch (Paszke et al., 2019)), this strategy still requires the entire model with the pruned parameters (indicated by the mask matrix) set as zero values for gradient computation during the backpropagation.

In our work, we propose ATTENDING, a novel FL approach integrated with personalized attentive pruning, aiming to reduce resource consumption of the heterogeneous clients meanwhile enhancing model performance on non-IID data. Additionally, we develop a specialized attention matching mechanism to aggregate the heterogeneous models resulting from personalized attentive pruning. We summarize our contributions as follows:

- We design a specific attention module to capture non-IID data features and assess the importance scores of model parameters on the clients thereby generating personalized compact local models for clients.
- We propose a specific aggregation algorithm integrated with an attention matching mechanism, enabling the aggregation of heterogeneous local models without the assistance of binary mask matrices.
- We implement and evaluate the proposed ATTENDING with a real FL platform. Experimental results on popular neural networks and benchmark datasets demonstrate that it outperforms baselines up to 11.3% while achieving a 32% reduction in model footprint.

## 2 RELATED WORK

**Federated Learning.** Federated learning enables collaborative training of complex models among distributed clients while keeping the local data on the client (McMahan et al., 2017; Zhou et al., 2022; Zhang et al., 2022; Jiang et al., 2023; Lu et al., 2023; Hu et al., 2024; Qiao et al., 2024; Jiang et al., 2024). As a classical FL approach, FedAvg is originally proposed by McMahan et al. (McMahan et al., 2017) to train and aggregate the local learning models. In each communication round of FedAvg, the local models are trained on the clients and aggregated on the central server. To view the systems and data heterogeneity in FL, Li et al. (Li et al., 2020) propose FedProx, which can be viewed as a generalization and re-parameterization of FedAvg in the non-IID data setting. Wang et al. (Wang et al., 2020a) demonstrate that the element-wise averaging of weights in FedAvg is a shortcoming due to the permutation invariance of neural network parameters, and proposes the FedMA to alleviate the detrimental effects caused by the permutation invariance. However, the above studies focus on improving model accuracy in FL or addressing the challenges posed by non-IID data, whereas neglecting the resource consumption of the clients. In contrast to the aforementioned studies, our proposed approach leverages the attention mechanism to mitigate the model accuracy degradation caused by non-IID data and reduce resource consumption through attentive model pruning.

**Model Compression in Federated Learning.** In FL, model compression techniques are used to reduce resource consumption by shrinking the footprint of models (Horvath et al., 2021; Wu et al., 2021; Li et al., 2021; Jiang et al., 2022a;b; Huang et al., 2024; Yi et al., 2024). Li et al. (Li et al., 2021) propose the Hermes, which addresses both data heterogeneity and communication efficiency issues in federated learning. Hermes leverages the structured pruning technique to find a smaller sub-model for each client, ensuring a more efficient training and communication process in FL. Yi et al. (Yi et al., 2024) propose FedP3, aiming to address model heterogeneity among clients

while enhancing the privacy of FL. FedP3 incorporates personalized network pruning techniques to optimize the performance and efficiency of local models. However, these studies necessitate binary mask matrices to indicate the model structure of the pruned local models during aggregation, which has been demonstrated to be adverse to FL (Wang et al., 2020a; Jiang et al., 2022a). Conversely, we propose an attentive model pruning algorithm to reduce the computational and storage overhead, alongside an attention matching-assisted aggregation algorithm to aggregate heterogeneous models. This approach effectively circumvents the requirement of binary mask matrices during aggregation.

## 3 DESIGN OF ATTENDING

The attention mechanism has proven effective in centralized deep learning paradigms (Sabour et al., 2017; Li et al., 2019; Liu et al., 2021; Ouyang et al., 2023). However, unlike centralized deep learning, model pruning-enabled FL approaches cannot directly benefit from the attention mechanism due to its inherently heterogeneous nature. In FL, model pruning on clients produces a variety of local models with different architectures and weights due to the heterogeneity challenges. Traditional element-wise aggregation fails to aggregate heterogeneous models with different architectures. Moreover, the aggregated global model suffers from performance deterioration due to the permutation invariance problem, even if the clients share homogeneous system capability and data distribution. Thus, in this section, we first introduce an dedicated attention module for ATTENDING, which is a key component to capture features on heterogeneous data and evaluate the importance scores of the model parameters. Subsequently, we propose attentive model pruning that leverages the importance scores generated by the attention module to prune local models, thereby producing personalized models for heterogeneous clients. To circumvent the adverse effects caused by the use of binary mask matrices during model aggregation, we propose a novel aggregation algorithm that aggregates heterogeneous models without relying on binary mask matrices, thereby mitigating the associated negative impacts.

### 3.1 ATTENTION MODULE FOR ATTENDING

**Design of Attention Module.** We leverage both spatial and channel attention mechanisms to construct the attention module. Spatial attention is particularly effective for enhancing local models to extract features from non-IID data, while channel attention is employed to assess the importance of model parameters at the channel level.

For spatial attention, we apply a grouping strategy to divide the channels of the feature map into $g$ groups to reduce the computation complexity and capture specific semantic responses.[1] Each group possesses a vector representation at every spatial position, exhibiting strong responses in critical regions (e.g., eye or nose regions in a dog's image), while displaying nearly zero vectors in other regions (e.g., non-meaningful background). For each group, the vector representation $\mathcal{F}_{loc}$ for each position $n \in N$ is denoted as $\mathcal{F}_{loc} = \{\mathcal{F}_1, \ldots, \mathcal{F}_n, \ldots \mathcal{F}_N\}$. Where $\mathcal{F}_n \in \mathbb{R}^{\frac{u}{g}}$ is the local feature at every position, $u$ represents the number of channels, and $N = h \times w$ with $h$ and $w$ represent the height and width of the feature map respectively. Considering the presence of unavoidable noise and similar patterns, we exploit the overall information of the whole group space to improve semantic feature learning in critical regions. The global feature vector $\mathcal{F}_{glo}^k$ of group $k \in [1, g]$ are calculated as follows:

$$\mathcal{F}_{glo}^k = concat(pool_a(\mathcal{F}_{loc[:u/2g,:,:]}^k), pool_m(\mathcal{F}_{loc[u/2g:,:,:]}^k)), \tag{1}$$

where $concat$ is concatenation operation, $pool_a$ is average pooling operation, $pool_m$ is maximum pooling operation. For convenient representation, we omit superscript $k$ in the subsequent description. The similarity between the global feature and local feature at each position is determined by:

$$\mathcal{S}_n = \mathcal{F}_{glo} \cdot \mathcal{F}_n, \tag{2}$$

and the similarity $\mathcal{S}_n$ is normalized as follows:

$$\hat{\mathcal{S}}_n = \frac{\mathcal{S}_n - \mu_{\mathcal{S}}}{\sigma_{\mathcal{S}} + \varepsilon}, \tag{3}$$

---

[1]It is worth noting that the grouping strategy is exclusively applicable to the "spatial attention" range. Once the "spatial attention" is computed, the channels will no longer be grouped.

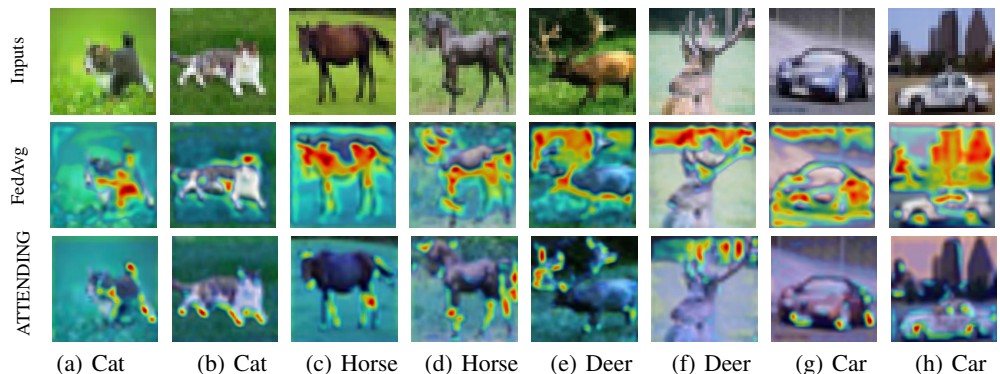

(a) Cat    (b) Cat    (c) Horse    (d) Horse    (e) Deer    (f) Deer    (g) Car    (h) Car

Figure 1: Visual explanations of FedAvg and ATTENDING utilizing Grad-CAM (Selvaraju et al., 2017). The top row is the original input images extracted from CIFAR10 dataset (with a resolution of $32 \times 32$ pixels). The middle row is the attention map generated by the global model trained with FedAvg. The bottom row is generated by the global model trained with ATTENDING.

where $\varepsilon$ is a constant added for numerical stability, $\mu_{\mathcal{S}}$ and $\sigma_{\mathcal{S}}$ are the mean and the standard deviation. Then, the spatial attention is calculated by the sigmoid function as follows:

$$\mathcal{A}_{\mathcal{S}} = sigmoid(\hat{\mathcal{S}}_n). \tag{4}$$

Following the calculation of spatial attention $\mathcal{A}_{\mathcal{S}}$, the channel attention $\mathcal{A}$ is derived through a combination of average pooling and maximum pooling operations as follows:

$$\mathcal{A} = sigmoid(GN(pool_a(\mathcal{A}_{\mathcal{S}})) + GN(pool_m(\mathcal{A}_{\mathcal{S}}))), \tag{5}$$

where $GN$ represents the group normalization operation (Wu & He, 2018). The channel attention $\mathcal{A}$ is the final attention obtained by the attention module. To enhance clarity, we further describe the process of forward propagation within the attention module in Appendix B.

It is worth noting that the inserted attention module consumes negligible client resources compared to the resource-intensive convolutional layers. For example, in the 2NN model used in our experiments, the attention module increases only 0.2% of the trainable model parameters.

**Visual Explanations.** We also provide visual explanations illustrating the effect of the attention mechanism in FL with non-IID data. Fig.1 offers an intuitive demonstration underscoring the effectiveness of the attention module on non-IID data. These visual explanations are generated by Grad-CAM (Selvaraju et al., 2017) using the optimized global model. A comparative analysis with the conventional FedAvg(McMahan et al., 2017) is also conducted, highlighting the superior performance of our proposed approach. As depicted in Fig. 1, ATTENDING exhibits more concentrated responses in critical regions, such as legs, tails, ears, deer horns, and automobile tires, while nearly zero responses in other regions, such as backgrounds. In contrast, the popular FedAvg, lacking an attention mechanism, struggles to focus on critical regions. Additionally, FedAvg often disperses attention to irrelevant backgrounds, as observed in the last four images in Fig. 1. Furthermore, the visual explanations in Fig. 1 provide evidence for the effect of the attention mechanism in model pruning, demonstrating that the proposed ATTENDING efficiently reduces the model footprint without significant performance loss on non-IID data.

## 3.2 Personalized Attentive Model Pruning

Based on the results of the attention module, we introduce a novel algorithm for personalized model pruning guided by attention scores. As shown in Fig. 2, each client trains the original model with attention modules (i.e., attentive training) and then conducts channel pruning guided by the attention scores to shrink the footprint of the model. Algorithm 1 presents the detailed process of attentive model pruning on each client.

---

**Algorithm 1** Attentive Model Pruning Algorithm

---

**Input:** Local model $\boldsymbol{\theta}_c$ on client $c$, channel set $E$ of target layers, attention modules with parameters $\boldsymbol{\theta}_a$, local data set $\mathcal{D}_c$, pruning ratio $p_c$
**Output:** Pruned local model with less parameters $\boldsymbol{\theta}'_c$
 1: Insert attention modules into the local model $\boldsymbol{\theta}_c$: $\boldsymbol{\theta}'_c \leftarrow \boldsymbol{\theta}_c \cup \boldsymbol{\theta}_a$
 2: **for** data batch $d \in \mathcal{D}_c$ **do**
 3:     Update $\boldsymbol{\theta}'_c$ on data batch $d$
 4:     Calculate attention $\mathcal{A}_c^d$ according to equation 5
 5: **end for**
 6: Calculate attention scores $\mathcal{M}_c$ according to equation 6
 7: Calculate attention threshold $\hat{m}$ according to equation 7
 8: **for** channel $e \in E$ with attention score $m_e$ **do**
 9:     **if** $m_e \leq \hat{m}$ **then**
10:         Remove the parameters $\boldsymbol{\theta}_e$ of channel $e$: $\boldsymbol{\theta}'_c \leftarrow \boldsymbol{\theta}'_c - \boldsymbol{\theta}_e$
11:     **end if**
12: **end for**
13: Remove the attention modules from local model $\boldsymbol{\theta}'_c$: $\boldsymbol{\theta}'_c \leftarrow \boldsymbol{\theta}'_c - \boldsymbol{\theta}_a$
14: **return** $\boldsymbol{\theta}'_c$

---

In lines 1-6, the client first initiates the incorporation of attention modules behind the target layers as a preparatory step. Subsequently, the client executes the mini-batch-based local training procedure on its local dataset. Within this local training process, the attention module computes attention scores for each data batch utilizing equation 5. Due to mini-batch-based training resulting in varying attention matrices across different data batches, it is crucial to mitigate the bias introduced by these batches. Consequently, we calculate the attention score matrix $\mathcal{M}_c$ for each local model as follows:

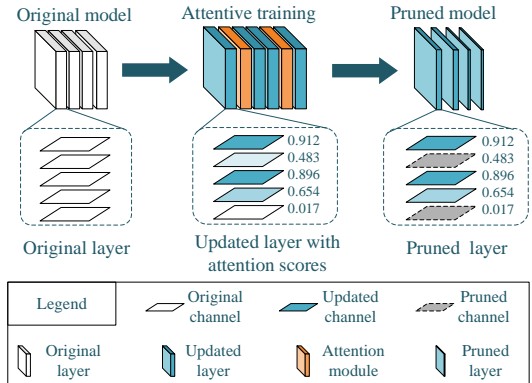

Original model     Attentive training     Pruned model

Original layer     Updated layer with     Pruned layer
                   attention scores

Legend    Original channel    Updated channel    Pruned channel

Original layer    Updated layer    Attention module    Pruned layer

Figure 2: An example of attention scores guided channel pruning on the client.

$$\mathcal{M}_c = \frac{|d|}{|\mathcal{D}_c|} \sum_{d \in \mathcal{D}_c} \mathcal{A}_c^d, \quad c \in \mathcal{C}, \qquad (6)$$

where $d$ is data batch in local dataset $\mathcal{D}_c$, $|d|$ is the number of data samples of $d$, and $\mathcal{A}_c^d$ is the attention scores matrix calculated on data batch $d$. Finally, the attention scores matrix $\mathcal{M}_c$ across the entirety of the local dataset $\mathcal{D}_c$ is computed via equation 6.

In lines 7-12, given a predefined pruning ratio $p_c$, channels with attention scores less than the attention threshold $\hat{m}$ are pruned to generate personalised local models.[2] Specifically, the attention threshold $\hat{m}$ is calculated as:

$$\hat{m} = \mathcal{M}_c(|\hat{E}|), \quad |\hat{E}| = |E| \times p_c, \qquad (7)$$

where $E$ is the channel set of all target layers, $|E|$ is the number of channels in $E$, and $\mathcal{M}_c(|\hat{E}|)$ denotes the $|\hat{E}|$-th smallest value in $\mathcal{M}_c$. Subsequently, ATTENDING remove the $|\hat{E}|$ channels with the smallest attention scores.

In line 13, the attention modules are removed to avoid unnecessary communication costs and reduce the resource consumption. In the subsequent communication rounds of FL, only the parameters in pruned local models will be transmitted between the clients and the central server.

---

[2]In this work, we follow the conventional assumption in resource-constrained FL that the system information of devices is available to the central server, and therefore the devices can negotiate with the central server to choose the appropriate pruning ratios.

Figure 3: Overview of the aggregation algorithm with attention matching in the first communication round (the legend can be seen in Fig. 2, and steps have been numbered as ❶-❼).

## 3.3 AGGREGATION OF HETEROGENEOUS MODELS

Fig. 3 shows an overview of the aggregation algorithm of ATTENDING in the first communication round of FL. For ease of representation, there are three clients (i.e., Device 1, Device 2, and Device 3) with different system capabilities participating in the FL process, and only one target layer containing 5 channels in the local model for each client.

Firstly, the clients download a original global model from the server (Step ❶) and update the weights of the local models with attention modules (Step ❷). Then they conduct personalized model pruning with different prune ratios via Algorithm 1. The personalized model pruning procedure on different clients will generate heterogeneous local models due to the various system capabilities.

Then, the local models are sent to the central server for aggregation (Step ❸). Due to the inherent heterogeneity among these local models, direct weighted averaging of these local models leads to a significant degradation in overall model performance. Inspired by the findings in existing work on model interpretability (Sabour et al., 2017; Li et al., 2019), channels with high scores play a crucial role in capturing specific semantic responses while those with low scores are susceptible to inherent noise and the presence of similar patterns. Thus we design a simple yet efficient *Attention Matching* mechanism to aggregate channels according to their attention scores, thereby enhancing the performance of the aggregated global model (Step ❹). Specifically, for each target layer, the attention matching mechanism reorganizes the channels, ensuring that the indexes of channels with small attention scores are positioned before those with larger attention scores. This arrangement is calculated as follows:

$$Index(t) < Index(q), \quad \forall m_t \leq m_q, \tag{8}$$

where the $Index$ function denotes the position of the respective channel within the target layer, $t$ and $q$ indicate different channels in the target layer with attention scores $m_t$ and $m_q$, respectively. In other words, the attention matching mechanism rearranges the channels within the target layer based on their attention scores, and therefore reduces the effects of permutation invariance.

Following that, the server reconstructs the structure of the pruned models using the original global model, ensuring that the local models align precisely with the structure of the global model (Step ❺). This alignment facilitates the application of weighted averaging to aggregate the local models.

Finally, the server aggregates recovered models through weighted averaging (Step ❻), and subsequently generates a personalized local model for each client (Step ❼). To elaborate, for each client $c$, the server prunes the last $p_c$ fraction of rearranged channels within the target layers of the aggregated model while retaining the remainder. The detailed aggregation algorithm for heterogeneous local models is provided in Appendix C.

In subsequent communication rounds, only the pruned local models are transmitted between the central server and the clients. The clients exclusively update the weights of the pruned models and send them directly to the central server without further pruning. Additionally, the central server abstains from executing attention matching as the order of the channels has already been established.

## 4 EXPERIMENTAL EVALUATION

In this section, we first present our experimental setup. Then, we compare the ATTENDING with SOTA methods in heterogeneous FL environments. Finally, ablation studies are conducted to scrutinize the effect of the attention module and pruning ratios, as well as the scalability of ATTENDING.

### 4.1 EXPERIMENTAL SETTING

**FL Environments.** We implemented ATTENDING with a benchmark FL platform FedML (He et al., 2020) and a popular deep learning framework PyTorch (Paszke et al., 2019). We build three FL environments to evaluate ATTENDING, as described in Table 1 (where "Env" represents "Environment"). The detailed structures of the 2NN and ResNet56 (He et al.,

Table 1: FL environments used in our experiments.

| Env | Client number | Model | Dataset | Train samples | Test samples |
|-----|------|------|------|------|------|
| Env-1 | 100 | 2NN | MNIST | 60,000 | 10,000 |
| Env-2 | 10 | ResNet56 | CIFAR10 and CIFAR100 | 50,000 | 10,000 |
| Env-3 | 1,000 | 2NN | MNIST | 60,000 | 10,000 |

2016) models are provided in Appendix D and the hyper-parameters utilized in model training are provided in Appendix E. Env-1 and Env-2 were adopted for evaluating learning performance, the effect of the attention module, the effect of pruning ratio, pruning ratios for the target layer (provided in Appendix H), and the effect of heterogeneity level of non-IID data (provided in Appendix I). Env-3 was adopted for evaluating the scalability. The experiments were performed on a GPU server with an Intel Core i9-10900K CPU and NVIDIA RTX3080Ti GPUs. Each experiment was executed three times using distinct random seeds, and the mean value was computed for analysis.

**Client Configuration.** We adopt the convention settings in FL (Wang et al., 2020b; Mei et al., 2022; Alam et al., 2022) to uniformly partition the set of clients $\mathcal{C}$ into five levels based on their system capabilities. All these clients are involved in the FL process in our experiments. We apply different pruning ratios for five levels of clients, where each level of clients is denoted as $\mathcal{C}_j, j \in [1, 5]$. The pruning ratio $p$ is configured as 0.7, 0.5, 0.3, 0.1, 0 for level 1 to level 5. A value of $p = 0$ indicates that clients at level 5 can train the complete model without any pruning applied.

**Datasets and Non-IID Partition.** We evaluate the performance of ATTENDING on the MNIST (LeCun et al., 1998), CIFAR10 (Krizhevsky et al., 2009), and CIFAR100 (Krizhevsky et al., 2009) datasets, for both IID and non-IID settings. In the IID setting, we uniformly sample an equal number of data samples for each client. In the non-IID setting, we adopt the Latent Dirichlet Allocation (LDA) (Wang et al., 2020a; Luo et al., 2021) strategy to partition the entire dataset among each client. The heterogeneity of data is determined by a concentration parameter $\alpha$. For the MNIST dataset, we set $\alpha$ to 0.1. For the CIFAR10 and CIFAR100 datasets, we set $\alpha$ to 0.5.

**Comparison Approaches.** We compare the proposed ATTENDING with 8 approaches: FedAvg (McMahan et al., 2017), FedDrop (Caldas et al., 2018), FedProx (Li et al., 2020), FedNova (Wang et al., 2020b), Hermes (Li et al., 2021), FedMP (Jiang et al., 2022b), FedGH (Yi et al., 2023), and FedP3(Yi et al., 2024). These approaches are either classical FL approaches or SOTA for addressing heterogeneity problems in resource-constrained edge environments. We provide a concise introduction to the comparison approaches in Appendix F.

### 4.2 COMPARISONS OF LEARNING PERFORMANCE

Table 2 and table 3 show the global model accuracy of FedAvg, FedDrop, FedProx, FedNova, Hermes, FedMP, FedGH, FedP3 and ATTENDING on the MNIST, CIFAR10, and CIFAR100 datasets during 100 communication rounds. As shown in Table 2 and Table 3, compared with Baseline, ATTENDING achieves 8.36% better average accuracy on the IID setting datasets and 11.3% better average accuracy in the non-IID setting datasets. Table 4 shows the pruning results of ATTENDING on 5 levels of clients. Where "Parameters" represents the number of parameters that determine storage resource consumption, and "FLOPs" are its floating point operations that determine computational resource consumption. As shown in Table 4, ATTENDING reduces 32% average footprints of both the 2NN and ResNet56 models.

Table 2: Accuracy comparison of Baseline, FedDrop, FedProx, FedNova, Hermes, FedMP, FedGH, FedP3, and `ATTENDING` on IID partitioning of MNIST, CIFAR10, and CIFAR100 datasets.

| FL Algorithm | Test accuracy (%) | | | |
|---|---|---|---|---|
| | MNIST | CIFAR10 | CIFAR100 | Average |
| Baseline (McMahan et al., 2017) | 95.50 (±0.15) | 79.85 (±0.92) | 38.76 (±0.79) | 71.37 (±0.62) |
| FedDrop (Caldas et al., 2018) | 96.59 (±0.13) | 80.46 (±0.20) | 39.92 (±0.23) | 72.32 (±0.19) |
| FedProx (Li et al., 2020) | 94.94 (±0.18) | 78.36 (±0.11) | 42.15 (±0.63) | 71.82 (±0.31) |
| FedNova (Wang et al., 2020b) | 95.98 (±0.21) | 77.87 (±0.13) | 42.15 (±0.70) | 72.00 (±0.35) |
| Hermes (Li et al., 2021) | 96.42 (±0.14) | 86.31 (±0.92) | 52.23 (±1.74) | 78.32 (±0.93) |
| FedMP (Jiang et al., 2022b) | 96.06 (±0.23) | 83.44 (±0.37) | 44.67 (±1.75) | 74.72 (±0.78) |
| FedGH (Yi et al., 2023) | **96.60** (±0.25) | 84.49 (±0.49) | 50.43 (±1.87) | 77.17 (±0.87) |
| FedP3 (Yi et al., 2024) | 95.84 (±0.24) | 87.30 (±0.97) | 48.35 (±1.39) | 77.16 (±0.87) |
| **ATTENDING (Ours)** | 96.22 (±0.20) | **88.45** (±0.30) | **54.51** (±1.39) | **79.73** (±0.63) |

Table 3: Accuracy comparison of Baseline, FedDrop, FedProx, FedNova, Hermes, FedMP, FedGH, FedP3, and `ATTENDING` on non-IID partitioning of MNIST, CIFAR10, and CIFAR100 datasets.

| FL Algorithm | Test accuracy (%) | | | |
|---|---|---|---|---|
| | MNIST | CIFAR10 | CIFAR100 | Average |
| Baseline (McMahan et al., 2017) | 84.07 (±0.97) | 63.62 (±0.50) | 33.57 (±1.94) | 60.42 (±1.14) |
| FedDrop (Caldas et al., 2018) | 88.02 (±2.50) | 65.96 (±2.16) | 32.55 (±0.74) | 62.18 (±1.80) |
| FedProx (Li et al., 2020) | 91.75 (±0.14) | 63.82 (±1.91) | 37.16 (±1.87) | 64.24 (±1.31) |
| FedNova (Wang et al., 2020b) | 89.46 (±0.98) | 70.10 (±1.66) | 38.60 (±1.98) | 66.05 (±1.54) |
| Hermes (Li et al., 2021) | 87.19 (±0.95) | 70.90 (±1.68) | 45.11 (±2.73) | 67.73 (±1.79) |
| FedMP (Jiang et al., 2022b) | 71.39 (±1.88) | 63.68 (±0.77) | 34.99 (±1.16) | 56.69 (±1.27) |
| FedGH (Yi et al., 2023) | 89.20 (±0.85) | 64.89 (±1.63) | 44.84 (±2.53) | 66.31 (±1.67) |
| FedP3 (Yi et al., 2024) | 87.55 (±0.57) | 67.75 (±0.76) | 43.95 (±1.84) | 66.42 (±1.06) |
| **ATTENDING (Ours)** | **93.59** (±0.25) | **73.00** (±1.89) | **48.57** (±1.87) | **71.72** (±1.34) |

Benefiting from the attentive pruning and aggregation algorithms, the proposed `ATTENDING` possesses the better capability of extracting features and avoids the adverse effects caused by element-wise aggregation, thus achieving better accuracy with less consumption of computational and storage resources. Additionally, to comprehensively investigate the learning process of comparison algorithms, we also provide detailed learning curves of FedAvg, FedDrop, FedProx, Fed-Nova, Hermes, FedMP, FedGH, FedP3 and `ATTENDING`, on the three datasets of both IID and non-IID settings in Appendix G.

Table 4: Pruning results of `ATTENDING` on clients.

| Level | Pruning ratio $p$ | 2NN model | | ResNet56 model | |
|---|---|---|---|---|---|
| | | Parameters | FLOPs | Parameters | FLOPs |
| 1 | 0.7 | 0.50M | 12.36M | 0.17M | 42.35M |
| 2 | 0.5 | 0.83M | 19.47M | 0.29M | 52.98M |
| 3 | 0.3 | 1.15M | 26.03M | 0.41M | 73.30M |
| 4 | 0.1 | 1.48M | 33.14M | 0.53M | 71.96M |
| 5 | 0 | 1.66M | 36.97M | 0.59M | 90.32M |
| Average | 0.32 | 1.12M | 25.59M | 0.40M | 66.18M |

### 4.3 EFFECT OF ATTENTION MODULE

The effect of the attention module is evaluated by removing the attention-related mechanism. Fig. 4 shows the performance of `ATTENDING` (i.e., `ATTENDING` w/ ATT) and its another version without the attention-related mechanism (i.e., `ATTENDING` w/o ATT). `ATTENDING` w/o ATT relies on the conventional L1-norm for executing model pruning on each client. In the model aggregation process, `ATTENDING` w/o ATT employs weighted averaging on local models without the attention matching mecha-

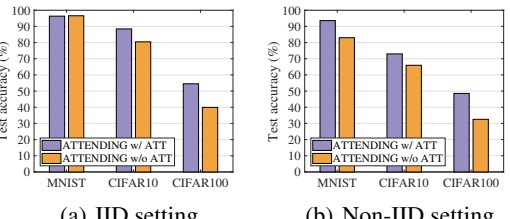

(a) IID setting  (b) Non-IID setting

Figure 4: Effect of attention module on global model accuracy in IID and non-IID settings.

nism as the attention scores are not present. In the IID setting, the accuracy of `ATTENDING` w/ ATT is 8% and 14.59% higher than `ATTENDING` w/o ATT on the CIFAR10 and CIFAR100 datasets, respectively, and comparable to it on the MNIST dataset. In the non-IID setting, the accuracy of `ATTENDING` w/ ATT surpasses that of `ATTENDING` w/o ATT on all three datasets. This result demonstrates that the attentive pruning and aggregation mechanism is effective for local models to extract features of both IID and non-IID data, particularly in intricate data scenarios.

## 4.4 EFFECT OF PRUNING RATIOS

The performance of the global model is closely related to the pruning ratio. In most cases, the smaller the pruning ratio of the model, the better the performance. Fig.5 shows the accuracy of the global model for different pruning ratios in both IID and non-IID settings. In the IID setting, the accuracy of the global model decreases as the pruning ratios increase on the CIFAR10 and CIFAR100 datasets. An interesting phenomenon is that the accuracy is highest at a pruning ratio of 0.5 on the MNIST dataset. The reason is that pruning the insignificant parts of

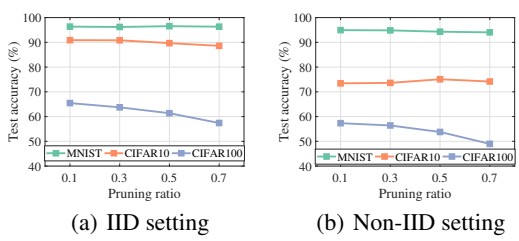

(a) IID setting  (b) Non-IID setting

Figure 5: Effect of different pruning ratios on MNIST, CIFAR10, and CIFAR100 datasets.

the model can reduce the possibility of overfitting and improve its generalization capability. In the non-IID setting, the accuracy decreases as the pruning ratios increase on the MNIST and CIFAR100 datasets. On the CIFAR10 dataset, the highest accuracy is also achieved with a pruning ratio of 0.5.

## 4.5 SCALABILITY OF ATTENDING

To investigate the scalability of `ATTENDING`, we compare its global model accuracy with that of four model pruning-based FL algorithms, FedDrop, Hermes, FedMP, FedGH, and FedP3 in a large-scale FL environment with a total of 1,000 clients. Fig. 6 shows the test accuracy on the MNIST dataset with different numbers of sampled clients participating in FL. For each communication round, only the sampled clients out of a total of 1,000 clients participate in the FL process. In the IID setting, a larger client sample rate results in relatively high accuracy.

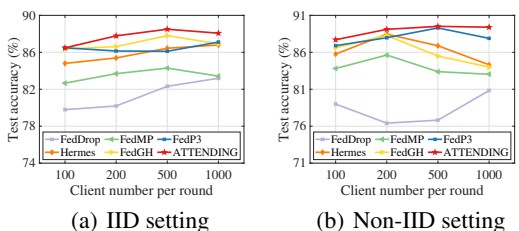

(a) IID setting  (b) Non-IID setting

Figure 6: Comparison of accuracy with IID and non-IID settings in large-scale FL environments.

The accuracy increases as the client sample number increases from 100 to 500, and decreases by 0.43% when the number increases to 1,000. In the non-IID setting, the accuracy reaches its highest point when the client sample number equals 200 and gradually decreases as the number increases from 200 to 1,000. The result of the scalability experiment demonstrates that `ATTENDING` is capable of being deployed in large-scale FL environments and still achieves satisfying performance.

## 5 CONCLUSIONS

In this paper, we have proposed a novel FL approach `ATTENDING`, enabling heterogeneous clients to participate in FL with personalized local models. We have introduced an attentive training and pruning algorithm for FL environments characterized by system and data heterogeneity. This algorithm aims to generate tailored local models while enhancing learning performance. To aggregate the local models, we have proposed a specific aggregation algorithm integrated with attention matching to address the model heterogeneity issue. We have implemented `ATTENDING` with a real FL platform and evaluated its performance. The experimental results on three benchmark datasets have demonstrated that `ATTENDING` significantly outperforms SOTA methods. Our future research endeavors will focus on exploring the implementation and extension of `ATTENDING` across various types of neural networks and multi-modality tasks, thereby enhancing its overall applicability.

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

# A  A Motivational Example

We present how system heterogeneity and data heterogeneity pose challenges to FL, thereby motivating the proposed design of `ATTENDING`.

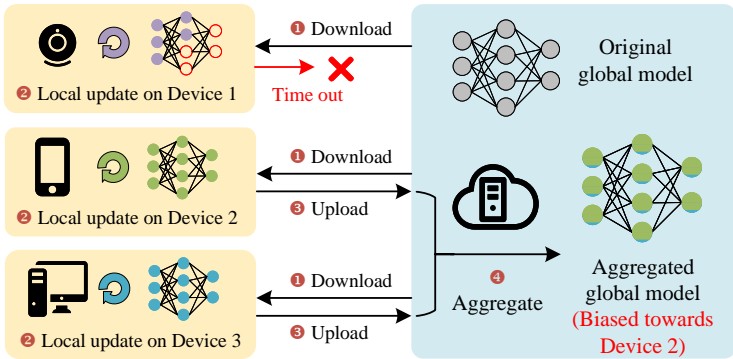

Figure 7: An example of FL process in an IoT network with heterogeneous devices (steps have been numbered as ❶-❹).

**System Heterogeneity.** In FL, the architectures of the local models on each client are usually the same as the global model, and system heterogeneity leads to the weaker clients' failure of local model updates within the maximum allowed time. For example, Fig. 7 describes a round of FL processing in an Internet of Things (IoT) network with three mobile devices. Device 1 encounters a failure in updating the model due to its stringent resource constraints. Consequently, the central server disregards the model information intended for updating by Device 1 and only aggregates the models updated from Device 2 and Device 3. This failure not only leads to the exclusion of Device 1 from the FL process but also hinders the central server from leveraging the data features extracted on Device 1.

In our work, the proposed approach addresses the system heterogeneity problem through a novel attentive pruning method. With attention-based model pruning, heterogeneous clients are able to update personalized local models with appropriate footprints, thus enabling all clients to participate in the FL process.

**Data Heterogeneity.** Different distributions of the heterogeneous local data pose another challenge on FL (Lim et al., 2020). Although the widely used FL approaches such as FedAvg (McMahan et al., 2017) can be applied on both IID and non-IID settings, the current work (Zhao et al., 2018; Li et al., 2021) demonstrates that FedAvg could be unstable or even diverge in the non-IID setting. According to (Zhao et al., 2018), the performance of a global model trained by the FedAvg (McMahan et al., 2017) has a 51% lower accuracy than a centrally-trained local model on the CIFAR10 dataset (Krizhevsky et al., 2009). For example, as shown in Fig. 7, the data distributions between Device 2 and Device 3 are distinct, resulting in diverse updated local models. Assuming that Device 2 possesses a larger number of data samples, the aggregated global model tends to be biased toward Device 2, thereby limiting the benefits attainable by Device 3.

In our work, the proposed approach utilizes the attention-based training technique to preserve the personality of local models and leverages attention matching to aggregate local models, thereby enhancing the model performance on non-IID data. Meanwhile, the proposed approach eliminates the need for binary mask matrices, thereby mitigating potential computation costs and preventing model accuracy deterioration.

# B  Process of Forward Propagation within The Attention Module

As shown in Fig. 8, for an input feature map $\mathcal{X}$, the spatial attention module first calculates its spatial attention $\mathcal{A}_{\mathcal{S}}$, yielding an intermediate result: $\mathcal{X} \odot \mathcal{A}_{\mathcal{S}}$. Then, the channel attention module calculates the channel attention $\mathcal{A}$ from the intermediate result. Finally, the output feature map can be calculated by: $\mathcal{X} \odot \mathcal{A}_{\mathcal{S}} \odot \mathcal{A}$. The channel attention, denoted as $\mathcal{A} \in \mathbb{R}^{u \times 1 \times 1}$, allows for the

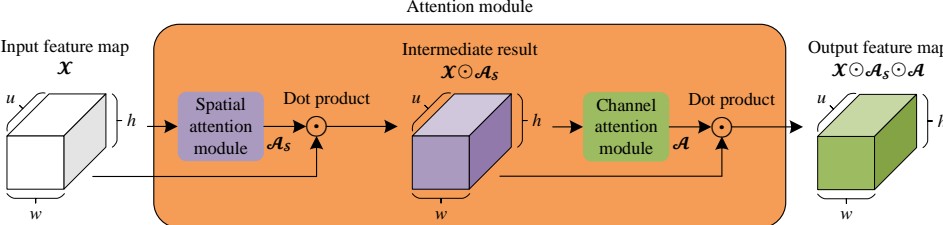

Figure 8: Process of forward propagation within the attention module.

direct extraction of attention scores for each channel. These attention scores will be used to prune channels.

## C  HETEROGENEOUS MODEL AGGREGATION ALGORITHM

Algorithm 2 presents the detailed aggregation process for heterogeneous local models.

---

**Algorithm 2** Heterogeneous Aggregation Algorithm

---

**Input:** Original global model with parameters $\Theta$, participating clients $c$, pruning ratio $p_c$, local data set $\mathcal{D}_c$.
**Output:** Aggregated global model with parameters $\Theta'$ and personalized local model $\theta'_c$
  **Client Updates($\Theta$):**
 1: $\theta_c \leftarrow$ Prune and update $\Theta$ by **Algorithm 1**
 2: Transmit the local update $\theta_c$ to the central server
  **Server Executes:**
 3: **for** each client $c \in \mathcal{C}$ **do**
 4:     Receive local models from clients:
      $\theta_c \leftarrow$ **Client Updates($\Theta$)**
 5:     Applying attention matching on $\theta_c$:
      $\theta_c \leftarrow Attention\ Matching(\theta_c)$
 6:     Recover model structure $\theta_c$ with $\Theta$:
      $\theta_c \leftarrow \theta_c \cup (\Theta - \theta_c)$
 7: **end for**
 8: Aggregate local models to generate the global model:
      $\Theta' \leftarrow \sum_{c=1}^{C} \frac{|\mathcal{D}_c|}{|\mathcal{D}|} \theta_c$
 9: **for** each client $c \in \mathcal{C}$ **do**
10:     $\theta'_c \leftarrow$ Prunes the last $p_c$ fraction of the channels
11: **end for**
12: **return**  $\Theta'$ and $\theta'_c$

---

## D  DETAILED MODEL STRUCTURES USED IN OUR EXPERIMENTS

The 2NN model used in our experiments is a shallow convolutional neural network. The original version of the 2NN model (McMahan et al., 2017) is adopted and its detailed structure is presented in Table 5. The ResNet56 model is a sophisticated neural network that features stacked residual blocks (He et al., 2016). In our experiments, we employed the same ResNet56 model as the one implemented in the FedML platform (He et al., 2020). The structure of our ResNet56 model is outlined in Table 6. The Residual layer used in the ResNet56 model is presented in Table 7. To enhance clarity and convenience, we have omitted the detailed structure of the Bottleneck, which is identical to its original implementation in (He et al., 2016).

Table 5: Model structure of the 2NN model.

| Index | Module | Type | Input shape | Output shape |
|---|---|---|---|---|
| 1 | conv1 | Conv | (1, 28, 28) | (32, 28, 28) |
| 2 | maxpool1 | MaxPooling | (32, 28, 28) | (32, 14, 14) |
| 3 | conv2 | Conv | (32, 14, 14) | (64, 14, 14) |
| 4 | maxpool2 | MaxPooling | (64, 14, 14) | (64, 7, 7) |
| 5 | flatten | Flatten | (64, 7, 7) | (3136) |
| 6 | linear1 | Dense | (3136) | (512) |
| 7 | linear2 | Dense | (512) | (10) |

Table 6: Model structure of the ResNet56 model.

| Index | Module | Type | Input shape | Output shape |
|---|---|---|---|---|
| 1 | conv1 | Conv | (3, 32, 32) | (16, 32, 32) |
| 2 | bn1 | BatchNorm | (16, 32, 32) | (16, 32, 32) |
| 3 | relu | ReLU | (16, 32, 32) | (16, 32, 32) |
| 4 | stage1 | Residual layer | (16, 32, 32) | (64, 32, 32) |
| 5 | stage2 | Residual layer | (64, 32, 32) | (128, 16, 16) |
| 6 | stage3 | Residual layer | (128, 16, 16) | (256, 8, 8) |
| 7 | avgpool | AvgPool | (256, 8, 8) | (256, 1, 1) |
| 8 | reshape | Reshape | (256, 1, 1) | (256) |
| 9 | linear | Dense | (256) | (10) for CIFAR10 (100) for CIFAR100 |

## E  TRAINING HYPER-PARAMETERS

To ensure a fair comparison with the SOTA approaches, we applied the same hyper-parameters for all approaches during model training. Table 8 shows a detailed description of these hyper-parameters.

Table 8: Training hyper-parameters adopted in FL process.

| Type | Hyper-parameter | Value |
|---|---|---|
| Global setting | Communication round | 100 |
| | Client number for MNIST | 100 |
| | Client number for CIFAR | 10 |
| | Client participation rate | 1 |
| | $\alpha$ of LDA for MNIST | 0.1 |
| | $\alpha$ of LDA for CIFAR | 0.5 |
| Local setting | Learning rate | 0.001 |
| | Weight decay | 0.001 |
| | Batch size | 512 |
| | Local epoch | 5 |

## F  COMPARISON ALGORITHMS

We introduce the comparison approaches utilized in our experimental section as follows:

**FedAvg** (McMahan et al., 2017): FedAvg is a pioneering work in FL and is widely used to aggregate local models. For model aggregation, FedAvg uploads the updated parameters of the local model to a central server for weighted averaging. FedAvg requires each client to update a local model with the same footprint as the global model and involves no resource optimization techniques, thus only the clients of level 5 (i.e., clients in $\mathcal{C}_5$) can participate in FedAvg.

Table 7: Structure of the "Residual layer" used in the ResNet56 model.

| Index | Module name | Module type | Input shape | Output shape |
|---|---|---|---|---|
| 1 | block1 | Bottleneck | (16, 32, 32) for stage1
(64, 32, 32) for stage2
(128, 16, 16) for stage3 | (16, 32, 32) for stage1
(128, 16, 16) for stage2
(256, 8, 8) for stage3 |
| 2 | downsample.conv | Conv2D | (16, 32, 32) for stage1
(64, 32, 32) for stage2
(128, 16, 16) for stage3 | (64, 32, 32) for stage1
(128, 16, 16) for stage2
(256, 8, 8) for stage3 |
| 3 | downsample.bn | BatchNorm2D | (64, 32, 32) for stage1
(128, 16, 16) for stage2
(256, 8, 8) for stage3 | (64, 32, 32) for stage1
(128, 16, 16) for stage2
(256, 8, 8) for stage3 |
| 4 | block2 | Bottleneck | (64, 32, 32) for stage1
(128, 16, 16) for stage2
(256, 8, 8) for stage3 | (64, 32, 32) for stage1
(128, 16, 16) for stage2
(256, 8, 8) for stage3 |
| 5 | block3 | Bottleneck | (64, 32, 32) for stage1
(128, 16, 16) for stage2
(256, 8, 8) for stage3 | (64, 32, 32) for stage1
(128, 16, 16) for stage2
(256, 8, 8) for stage3 |
| 6 | block4 | Bottleneck | (64, 32, 32) for stage1
(128, 16, 16) for stage2
(256, 8, 8) for stage3 | (64, 32, 32) for stage1
(128, 16, 16) for stage2
(256, 8, 8) for stage3 |
| 7 | block5 | Bottleneck | (64, 32, 32) for stage1
(128, 16, 16) for stage2
(256, 8, 8) for stage3 | (64, 32, 32) for stage1
(128, 16, 16) for stage2
(256, 8, 8) for stage3 |
| 8 | block6 | Bottleneck | (64, 32, 32) for stage1
(128, 16, 16) for stage2
(256, 8, 8) for stage3 | (64, 32, 32) for stage1
(128, 16, 16) for stage2
(256, 8, 8) for stage3 |

**Federated Dropout (FedDrop)** (Caldas et al., 2018): FedDrop integrates lossy compression techniques into the FL process. By generating a uniform, compact local model for all clients, FedDrop effectively mitigates the computational burden associated with local training and the corresponding communication costs. It is essential that this compact local model adheres to the resource constraints of the least capable clients, denoted as $\mathcal{C}_1$.

**FedProx** (Li et al., 2020): FedProx introduces a proximal term into the objective function, aiming to encourage clients to maintain similarity with the global model during local training, thereby allowing low-end clients to execute fewer local updates.

**FedNova** (Wang et al., 2020b): FedNova adopts dynamic learning rate adjustment and adaptive aggregation mechanisms to handle the heterogeneity among participants. FedNova adjusts each participant's influence on the model by using their contribution.

**Hermes** (Li et al., 2021): Hermes adopts structured pruning to find personalized sub-models for clients. The clients are responsible for updating the personalized sub-models, which are subsequently transmitted to the server for intersection-based averaging.

**FedMP** (Jiang et al., 2022b): FedMP prunes local models adaptively in each FL communication round to satisfy the heterogeneous resource limitations of clients. Meanwhile, it uses a R2SP scheme to aggregate heterogeneous models in different clients.

**FedGH** (Yi et al., 2023): FedGH enables the extraction of representations from local data by heterogeneous sub-models on clients. To facilitate knowledge transfer among clients, FedGH employs a shareable global header optimized using these representations to make predictions. We categorize FedGH as a method related to model pruning in our work as it utilizes sub-models with reduced width as heterogeneous local representation extractors.

**FedP3** (Yi et al., 2024): FedP3 involves dynamic network pruning, where each client trains a subset of the global model and sends pruned weights back to the server for aggregation. Through dynamic pruning and training, FedP3 is able to better address system heterogeneity with tailored models.

## G    LEARNING CURVES

We provide detailed learning curves of ATTENDING and FedAvg (McMahan et al., 2017), Fed-Drop (Caldas et al., 2018), FedProx (Li et al., 2020), FedNova (Wang et al., 2020b), Hermes (Li et al., 2021), FedMP (Jiang et al., 2022b), FedGH (Jiang et al., 2022b), and FedP3 (Yi et al., 2024) on three datasets of both IID setting and non-IID setting in Fig. 9 and Fig. 10. Specifically, Fig. 9 shows the results of ATTENDING, FedAvg, and optimization-based comparison approaches (i.e., FedProx and FedNova). Fig. 10 shows the results of ATTENDING and model pruning-based comparison approaches (i.e., FedDrop, Hermes, FedMP, FedGH, and FedP3). In Fig. 9 and Fig. 10, the "Acc_IID" and "Loss_IID" represent test accuracy and loss in the IID setting during 100 communication rounds, respectively. The "Acc_non-IID" and "Loss_non-IID" represent test accuracy and loss in the non-IID setting.

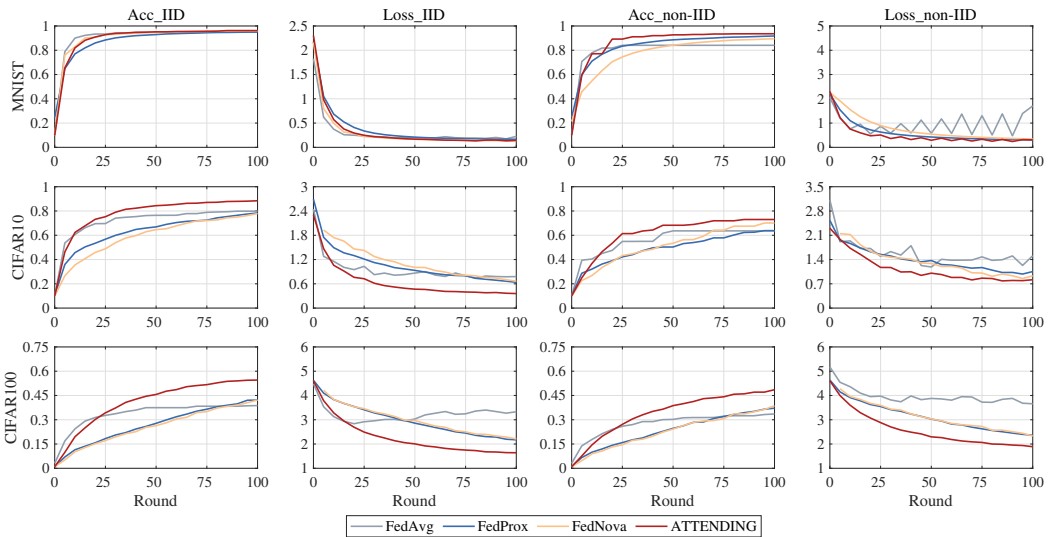

Figure 9: Test accuracy and test loss of FedAvg, FedProx, FedNova, and ATTENDING on three datasets of both IID setting and non-IID setting.

## H    PRUNING RATIOS FOR TARGET LAYERS

Given a pruning ratio $p$ for model pruning, we employed the constant pruning ratio $p$ for each target layer $l \in L$ in the 2NN model. For the ResNet56 model, we used diverse pruning ratios $p^l$ for target layers while ensuring that the pruning ratio $p$ for the entire model remained unchanged.

As discussed in (Yosinski et al., 2014; Donahue et al., 2014), the early layers of a neural network primarily contribute to its generality, while the later layers tend to dominate specificity. In the context of FL environments, the generality of local models on clients often carries greater significance than their specificity due to the distributed nature of FL. Specifically, each client in an FL environment must collaborate with other clients to optimize a global model, making local models with greater generality crucial for enhancing the performance of the global model. Additionally, FL often encounters performance degradation especially in non-IID settings, primarily because local models are trained specifically on their own local data and struggle to generalize to data from other clients.

Thus, we set small pruning ratios for the early layers of ResNet56, and large pruning ratios for its later layers to improve its generality. In order to reduce complexity, we establish distinct pruning ratios for individual stages of ResNet56, as opposed to targeting specific layers. These ratios are

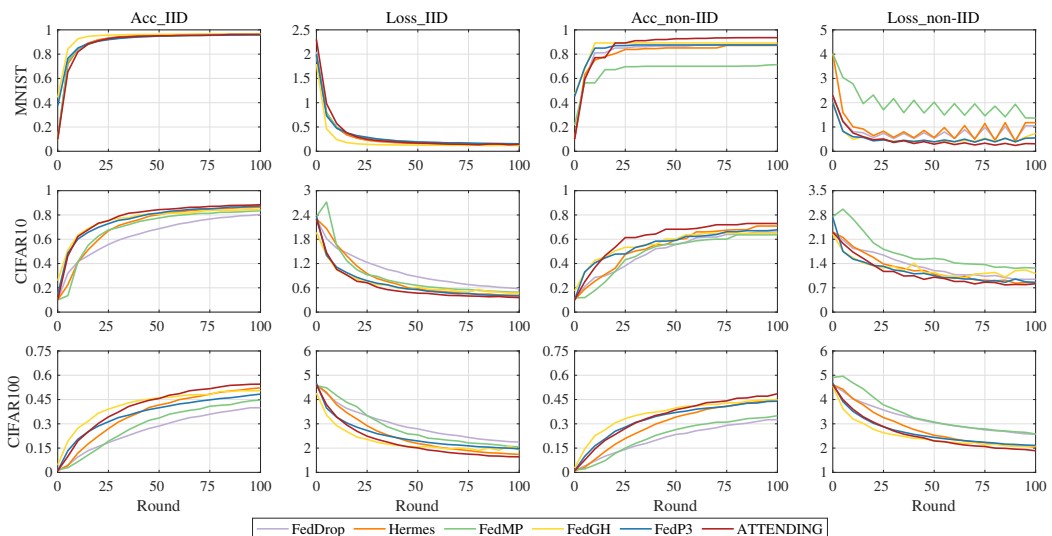

Figure 10: Test accuracy and test loss of FedDrop, Hermes, FedMP, FedGH, FedP3, and ATTENDING on three datasets of both IID setting and non-IID setting.

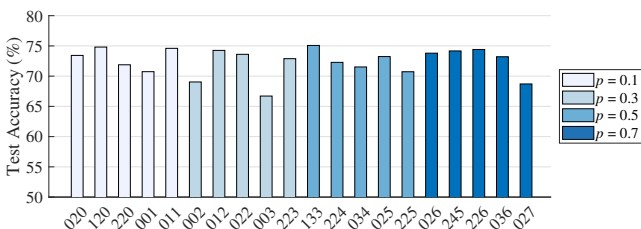

Figure 11: Performance of ATTENDING on non-IID setting of CIFAR10 dataset under various pruning ratios $p_{si}$.

formally denoted as $\widetilde{p}_s$, where $s$ represents the $s$-th stage of the ResNet56 model. To enhance the generality of the model under a given model pruning ratio, we ensure that:

$$\widetilde{p}_i \leq \widetilde{p}_j, \quad \forall i < j, \quad i, j \in S, \tag{9}$$

where $S$ denotes the set of indices for each stage in ResNet56. Given the inherent challenge in achieving a precise reduction of the model footprint to a specified value $p$ through structured model pruning, we select the pruning ratio $\widetilde{p}_s$ of the stages as far as possible to ensure the entire pruning ratio approx to $p$, which can be formulated as follows:

$$p \approx \frac{1}{|S|} \sum_{s=1}^{|S|} \widetilde{p}_s. \tag{10}$$

We designed 20 pruning strategies that satisfy equation 9 and equation 10 to investigate the impact of different pruning ratios $\widetilde{p}_s$ on model performance. We conduct this experiment with the ResNet56 model and non-IID setting of the CIFAR10 dataset. Fig. 11 shows the results, where the indexes of abscissa represent the pruning strategies for each stage of ResNet56. For example, "245" represents a pruning strategy with $\widetilde{p}_1 = 0.2$, $\widetilde{p}_2 = 0.4$, and $\widetilde{p}_3 = 0.5$, respectively. Note that when $p = 0.1$, equation 9 is not applied because the early layers of ResNet56 contain a small fraction of parameters whereas its later layers contain a large one.

In this paper, we choose the pruning strategies "120", "012", "133", and "226" corresponding to the pruning ratios $p$ of 0.1, 0.3, 0.5, and 0.7, respectively.

# I EFFECT OF HETEROGENEITY LEVEL OF NON-IID DATA

We evaluate the proposed `ATTENDING` with various data heterogeneity levels and study how they affect the model accuracy. The concentration parameter $\alpha$ in the LDA strategy controls the data heterogeneity level. The results of `ATTENDING` on the MNIST dataset and CIFAR10 dataset under different concentration parameters $\alpha$ are depicted in Fig. 12. The 2NN model is trained on the MNIST dataset, while the ResNet model is trained on the CIFAR10 dataset. As illustrated in Fig. 12, the efficacy of the `ATTENDING` method remains resilient amidst variations on the MNIST dataset, exhibiting a slight decline as $\alpha$ decreases. This phenomenon is attributed to the inherent simplicity of the classification task on the MNIST dataset, which experiences comparatively modest accuracy deterioration. In contrast, the CIFAR10 dataset, being more challenging, inevitably leads to larger accuracy degradation as $\alpha$ decreases.

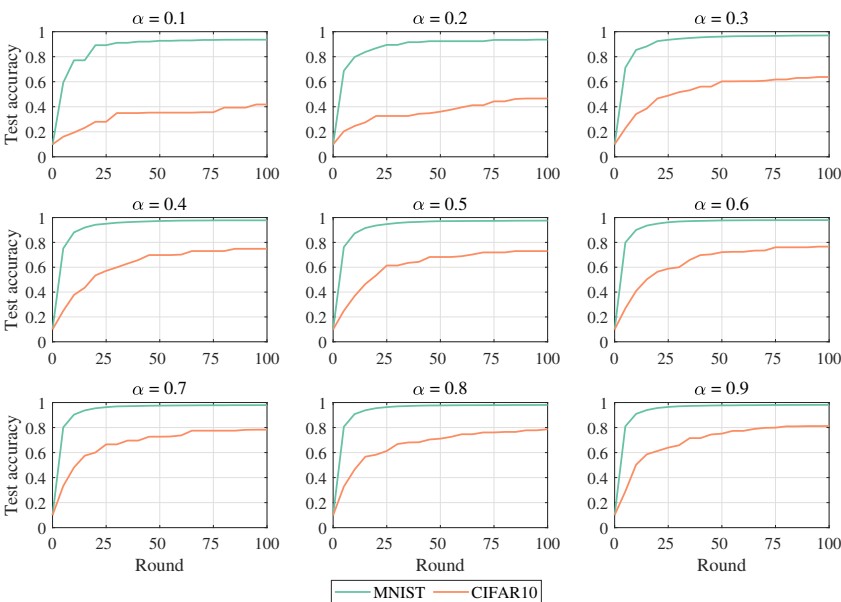

Figure 12: Test accuracy of `ATTENDING` with concentration parameters $\alpha$ ranging from 0.1 to 0.9 on the MNIST dataset and the CIFAR10 dataset.

