# OpenReview forum: "ATTENDING: Federated Learning with Personalized Attentive Pruning for Heterogeneous Clients"
_ICLR.cc/2025/Conference — ICLR 2025 Conference Withdrawn Submission_

### Official Review · Reviewer_FsjL · 2024-10-30

**Soundness:** 2
**Presentation:** 2
**Contribution:** 2
**Rating:** 3
**Confidence:** 4

**Summary:**

To address challenges from system and data heterogeneity in FL, the paper proposes ATTENDING, a personalized attentive pruning-enabled federated learning approach. ATTENDING enhances learning performance using an attention module and generates personalized local models through attentive pruning. Evaluation results demonstrate that ATTENDING outperforms baselines by up to 11.3% and reduces average model footprints by 32%.

**Strengths:**

1. The paper introduces the attention mechanism,  incorporating spatial and channel attention to effectively addresses both heterogeneity in FL.
2. The paper clearly illustrate the proposed method, including the attention module design, the pruning algorithm, and the aggregation mechanism.
3. The paper conducts extensive experiments across various datasets and settings, demonstrating the robustness and scalability of ATTENDING.

**Weaknesses:**

1. To obtain the pruned client models, it is necessary for all client models to initially possess sufficient resources to run the original models. This requirement appears to contradict the system heterogeneity challenge that ATTENDING aims to address.

2. In line 125, could the authors please clarify that what is the permutation invariance problem in Federated Learning (FL)?

3. The proposed attention module is designed for each client individually; however, these attention modules do not seem to have any alignments. How can it be ensured that the attention module "is a key component to capture features on heterogeneous data," as stated in line 127? It seems that the attenion module can only extract features from the local dataset, rather than from the heterogeneous data in the FL training system.

4. In Figure 3, it appears that the channels are aggregated based on the scores rather than their original positions in each layer. Could this have any impact, considering that channel weights from different positions may extract different features from the images?

5. In line 321, does this imply that only the first round involves pruning the client models? How can it be ensured that the initial model with the attention module can extract appropriate attentions when the model weights are initialized randomly? Additionally, what would be the results if the channels with the highest scores were pruned, or if pruning were done randomly?

6. It would be beneficial to include additional results that why ATTENDING performs better , such as the distribution of attention values for different channels and the effect of different attention scores.

7. Could ATTENDING be applied in Transformers? If so, what modifications would be necessary to adapt the approach for use with Transformers?

**Questions:**

Please see the weaknesses.

---

> ### Author Response · Authors · 2024-11-23
>
> We sincerely thank you for your constructive and helpful comments. Below we address your concerns in order. The discussion here will be properly incorporated into a new version of our paper.
>
> > Weakness 1: To obtain the pruned client models, it is necessary for all client models to initially possess sufficient resources to run the original models. This requirement appears to contradict the system heterogeneity challenge that ATTENDING aims to address.
>
> **Response:** It is correct that in the first round of FL, all clients initialize with the original model. However, the pruning process is designed to address the system heterogeneity challenge by tailoring models to the specific resource constraints of each client after the initial round. This means that the full model is required only once at the start of training.
> The system heterogeneity challenge primarily concerns the ongoing computational and storage requirements for clients during training. Once pruning is applied in ATTENDING, each client trains a personalized, pruned model that fits within its resource constraints. This significantly reduces the computational and storage burden for subsequent rounds.
>
> > Weakness 2: In line 125, could the authors please clarify that what is the permutation invariance problem in Federated Learning (FL)?
>
> **Response:** We thank the reviewer for requesting clarification on the permutation invariance problem in Federated Learning (FL), as mentioned in line 125. Below, we provide a detailed explanation:
>
> Permutation invariance refers to the property that the arrangement (or ordering) of parameters in neural network layers does not affect the functionality of the model. For example, in a fully connected layer, permuting the neurons (and correspondingly their weights and biases) results in a mathematically equivalent model. However, in FL, this property can lead to challenges during model aggregation when local models are heterogeneous due to personalized training or pruning.
>
> > Weakness 3: These attention modules do not seem to have any alignments. How can it be ensured that the attention module "is a key component to capture features on heterogeneous data,"? It seems that the attenion module can only extract features from the local dataset, rather than from the heterogeneous data in the FL training system.
>
> **Response:** Alignment of Attention Scores: Although the attention modules are trained locally, their outputs (e.g., attention scores) are used to align channels during the attention-matching aggregation mechanism, ensuring that the global model benefits from the most significant features learned across clients. This alignment facilitates knowledge transfer from diverse local data distributions to the global model.
>
> Adaptive Feature Extraction on Heterogeneous Data: In this work, data heterogeneity refers to the non-IID nature of data between different clients. The attention modules are indeed designed and trained locally on each client to adapt to the unique data distribution of that client. This design ensures that the attention mechanism captures features that are most relevant to the client’s local dataset. While the attention modules primarily operate on local data, they play a critical role in improving the personalized performance of client models, which is essential in heterogeneous FL settings.
>
> > Weakness 4: In Figure 3, it appears that the channels are aggregated based on the scores. Could this have any impact, considering that channel weights from different positions may extract different features from the images?
>
> **Response:** Due to the existence of permutation invariance in neural networks, the sequences of channels are inherently various in distinct local models. Even if we directly aggregate the local models without rearranging them, the misalignment still happens.
> Thus, we proposed an attention-matching mechanism, aiming to alleviate such misalignment. The attention-matching mechanism rearranges the channels within the target layer based on their attention scores, ensuring that the channel order is determined by the attention scores and therefore reduces the effects of permutation invariance. The rationale behind averaging channels with high scores from different local models is rooted in their role as key contributors to capturing specific semantic responses. Conversely, averaging channels with low scores is driven by the understanding that these channels are susceptible to inherent noise and the presence of similar patterns, which can dilute their significance.

---

> ### Author Response · Authors · 2024-11-23
>
> > Weakness 5: In line 321, does this imply that only the first round involves pruning the client models? How can it be ensured that the initial model with the attention module can extract appropriate attentions when the model weights are initialized randomly? Additionally, what would be the results if the channels with the highest scores were pruned, or if pruning were done randomly?
>
> **Response:** Yes, as stated in line 321, the first round involves pruning the client models based on the initial attention scores. We adopt the first round (5 epochs) of local training to determine the pruned models for two reasons: on one hand, pioneering work related to model design and neural architecture search (NAS) [1,2] have demonstrated that few epochs are sufficient to obtain a coarse estimate on a sub-model. For example, MnasNet [1] adopted 5 epochs as a training step to perform architecture searching. On the other hand, the local model of each client will be updated separately only in the first round of communication of the FL, whereas after the first round of communication, the local model will be aggregated and will no longer capture the features of each client specifically. Thus, the pruning operation is performed in the first communication round before the model aggregation.
>
> Pruning Channels with the Highest Scores: If channels with the highest attention scores were pruned instead of the lowest, the model’s capacity to extract key features would be compromised, leading to a significant drop in performance. Attention scores reflect the importance of channels; thus, removing the most important ones would undermine the model’s ability to learn effectively.
>
> Random Pruning: Pruning channels randomly would likely result in suboptimal model performance, as it does not account for the varying importance of different channels. Our experimental results (in Section 4.3, Table 2 and Table 3) show that attention-guided pruning significantly outperforms random pruning (i.e., FedDrop) by ensuring that less important channels are removed while retaining critical ones.
>
> *[1] Tan, Mingxing, et al. "Mnasnet: Platform-aware neural architecture search for mobile." Proceedings of the IEEE/CVF conference on computer vision and pattern recognition. 2019.
> [2] Howard, Andrew, et al. "Searching for mobilenetv3." Proceedings of the IEEE/CVF international conference on computer vision. 2019.*
>
> > Weakness 6: It would be beneficial to include additional results that why ATTENDING performs better.
>
> **Response:** We would like to highlight that, in Figure 1, we visualize the attention maps generated by the attention module, which directly reflect the relative importance of different features for each client. These maps show how the attention mechanism selectively focuses on important features, which are then preserved through the pruning and aggregation steps. This provides a clear and intuitive understanding of how ATTENDING prioritizes critical features from heterogeneous data. Specifically, the attention maps highlight that, by focusing on the most relevant features, ATTENDING effectively retains the information necessary for accurate predictions while pruning less important features, which leads to better performance in heterogeneous FL settings compared to traditional methods that do not adapt pruning based on feature importance.
>
> > Weakness 7: Could ATTENDING be applied in Transformers? If so, what modifications would be necessary to adapt the approach for use with Transformers?
>
> **Response:** We appreciate the reviewer’s insightful question regarding the applicability of ATTENDING to Transformer architectures. Unfortunately, ATTENDING, as currently designed, is not directly applicable to Transformers due to fundamental differences in architecture and attention mechanisms. We recognize the potential benefit of extending attention-guided pruning techniques to Transformer architectures in federated learning settings. Our future research could explore how similar principles could be applied to the pruning of attention heads or layers in Transformers. Such an extension would require rethinking the attention module and aggregation mechanism to account for the unique structure of Transformer models.

---

### Official Review · Reviewer_bZhL · 2024-10-30

**Soundness:** 3
**Presentation:** 3
**Contribution:** 2
**Rating:** 5
**Confidence:** 4

**Summary:**

The paper proposes an attention-based pruning scheme to address the heterogeneity problems in FL. The attention scheme includes both channel attention and spatial attention to calculate a per-channel score for pruning, instead of using mask, which according to the authors, provides better robustness to performance permutation and reduced computation cost. The algorithm is implemented in FedML and compared with several SoTA methods showing reasonable benefits in terms of accuracy and cost.

**Strengths:**

The idea is somewhat novel in the sense that there aren't many pruning works in FL setups that employ attention-based scores, mostly would, like the authors mentioned, used masks.
The writing and description are in general clear and easy to follow.
The algorithm seems reasonable to function, as proved by the evaluation, which is quite sufficient.
Some details are well explained, e.g., the effect of pruning ratios on different layers in Appendix H.

**Weaknesses:**

1. The paper's claim on the weakness of existing mask-based pruning approaches only apply to some of the existing works. Those with different approaches, e.g., using sign supermask instead of binary supermask, or, conduct pruning in the server instead in the clients [1], do not have the mentioned weakness yet seem to be overlooked in the work.
2. The attention module seems quite similar, if not identical, to the classical CBAM [2], please clarify the innovation there.
3. In 3.3, the attention-matching mechanism rearranges the channels based on attention scores before aggregation. I might have misunderstood this, but wouldn't this operation cause chaos since the sequence of channels is now changed and thus channels learning different features are aggregated, e.g., those learing "edge" aggregated with those learning "texture"?

[1] HideNseek: Federated Lottery Ticket via Server-side Pruning and Sign Supermask, arXiv:2206.04385

[2] CBAM: Convolutional Block Attention Module, ECCV 2018

**Questions:**

As mentioned in the weakness,
1. What's the advantage of this work over those pruning works using sign-supermask server-side pruning, e.g., [1]?
2. What's the innovation/difference between this attention module and CBAM's [2]?
3. Wouldn't the attention-matching mechanism cause chaos since the sequence of channels is now changed and thus channels learning different features are aggregated?

[1] HideNseek: Federated Lottery Ticket via Server-side Pruning and Sign Supermask, arXiv:2206.04385

[2] CBAM: Convolutional Block Attention Module, ECCV 2018

---

> ### Author Response · Authors · 2024-11-23
>
> We sincerely thank you for your constructive and helpful comments. Below we address your concerns in order. The discussion here will be properly incorporated into a new version of our paper.
>
> > Weakness 1 \& Question 1: The paper's claim on the weakness of existing mask-based pruning approaches only apply to some of the existing works. Those with different approaches, e.g., using sign supermask instead of binary supermask, or, conduct pruning in the server instead in the clients [1], do not have the mentioned weakness yet seem to be overlooked in the work.
> > What's the advantage of this work over those pruning works using sign-supermask server-side pruning, e.g., [1]?
>
> **Response:** While different pruning approaches, such as those using sign supermasks or performing pruning on the server instead of clients, avoid the use of binary supermasks, the model performance of these methods is severely degraded due to the limited degree of personalization, especially on non-IID data. As demonstrated in prior work and supported by the empirical results in [1], these methods often suffer from significant performance degradation due to limited personalization, particularly in non-IID data settings.
>
> Additionally, we included a server-side pruning method, FedDrop, in the experimental section to highlight the performance differences between server-side and client-side pruning approaches. The results further demonstrate the advantages of client-side pruning, particularly in addressing client-specific heterogeneity.
>
> On the other hand, binary supermask-based methods have been shown to be more efficient than sign supermask-based methods. As noted in [1]: "...in the future, we will explore the efficacy brought by employing a binary optimizer [Helwegen et al., 2019] that only modifies signs of weights without the need for latent parameters like the sign supermasks." This evidence suggests that binary supermasks remain a practical and efficient choice for pruning in federated learning scenarios.
>
> *[1] HideNseek: Federated Lottery Ticket via Server-side Pruning and Sign Supermask, arXiv:2206.04385*
>
> > Weakness 2 \& Question 2: The attention module seems quite similar, if not identical, to the classical CBAM [2].
> > What's the innovation/difference between this attention module and CBAM's [2]?
>
> **Response:** While our attention module adopts the general concept of spatial and channel attention, there are critical distinctions in its design and application:
>
> Adaptation for FL: CBAM was originally designed for centralized deep learning tasks, where models have access to globally available data. In contrast, our attention module is specifically tailored for FL, addressing challenges such as non-IID data distributions and heterogeneous client capabilities.
>
> Grouping Strategy in Spatial Attention: Our spatial attention mechanism introduces a grouping strategy to divide feature map channels into multiple groups, reducing computational complexity and enhancing feature extraction for non-IID data. This grouping strategy is particularly effective in the resource-constrained environment of FL, where computational efficiency is critical.
>
> Attention-Driven Pruning and Aggregation: More importantly, unlike CBAM, which solely focuses on enhancing feature representations, our attention scores are directly used to guide the pruning process and aggregation mechanism. This integration makes the attention module a central component in addressing both data and system heterogeneity in FL.
>
> *[2] CBAM: Convolutional Block Attention Module, ECCV 2018*
>
> > Weakness 3 \& Question 3:  In 3.3, the attention-matching mechanism rearranges the channels based on attention scores before aggregation. I might have misunderstood this, but wouldn't this operation cause channels learning different features are aggregated, e.g., those learing "edge" aggregated with those learning "texture"?
>
> **Response:** Due to the existence of permutation invariance in neural networks, the sequences of channels are inherently various in distinct local models. Even if we directly aggregate the local models without rearranging them, the misalignment (e.g., channels learning "edge" features being aggregated with those learning "texture" features) still happens. Thus, we proposed an attention-matching mechanism, aiming to alleviate such misalignment. The attention-matching mechanism rearranges the channels within the target layer based on their attention scores, ensuring that the channel order is determined by the attention scores and therefore reduces the effects of permutation invariance. The rationale behind averaging channels with high scores from different local models is rooted in their role as key contributors to capturing specific semantic responses. Conversely, averaging channels with low scores is driven by the understanding that these channels are susceptible to inherent noise and the presence of similar patterns, which can dilute their significance.

---

### Official Review · Reviewer_Dmg9 · 2024-11-02

**Soundness:** 2
**Presentation:** 3
**Contribution:** 2
**Rating:** 5
**Confidence:** 3

**Summary:**

This paper proposes an attention pruning method called ATTENDING for federated learning to address the data heterogeneity and system heterogeneity issues. Specifically, spatial and channel attention are designed to extract features from Non-IID data and assess model parameters' importance, respectively. Moreover, each client executes model pruning, and the server executes attention aggregation based on attention scores to shrink the footprint. ATTENDING is evaluated on three datasets and results show that ATTENDING can improve test accuracy and reduce model footprint.

**Strengths:**

1. The paper is well-written, and the idea is easy to follow.

2. There are figures to help understand the methods.

**Weaknesses:**

1. Authors claim that one main advantage of ATTENDING compared with other methods is reducing communicating, computational, and storage overhead. However, there is limited theoretical analysis regarding these perspectives. One example of computation (lines 193-196) is insufficient.

2. The authors also claim that their method does not need binary mask matrices compared with other pruning methods. However, the disadvantages of using matrices and the advantages of ATTENDING are not well analyzed and compared theoretically and experimentally.

3. The experiment part mainly focuses on accuracy, also lacks communication and storage results. How many computation and communication resources do ATTENDING and baselines require? Some results tables or figures should be included to illustrate such comparisons. This can help support what the authors claim.

4. MNIST is used in two of the three Envs, but this dataset is too simple, and more complicated datasets are suggested to replace it.

5. ATTENDING is designed to address the data heterogeneity and system heterogeneity issues, as claimed by the authors. However, related experimental settings are not so detailed and heterogenous. Experiments should include more different alpha values regarding data heterogeneity and more different client configurations regarding system heterogeneity issues.

6. Figure 1 only compares ATTENDING with FedAvg; other baselines aiming for the data heterogeneity issue should also be included, and I believe such a comparison is more valuable and fair.

7. The limitations of ATTENDING should also be discussed in the paper, except for the advantages.

**Questions:**

What is the intuition to calculate the threshold in the current way, i.e., equation (7)?

How many computation and communication resources do ATTENDING and baselines require?

---

> ### Author Response · Authors · 2024-11-23
>
> We sincerely thank you for your constructive and helpful comments. Below we address your concerns in order. The discussion here will be properly incorporated into a new version of our paper.
>
> > Weakness 1: Authors claim that one main advantage of ATTENDING is reducing communicating, computational, and storage overhead. However, there is limited theoretical analysis regarding these perspectives. One example of computation (lines 193-196) is insufficient.
>
> **Response:** We demonstrated the reduction in overhead through extensive experiments:
> The proposed personalized pruning significantly reduces the number of parameters and FLOPs for local models, as shown in Table 4. This reduction directly translates to lighter computational requirements for resource-constrained clients. By pruning model parameters, ATTENDING decreases the average model size by 32%, which is particularly beneficial for devices with limited storage capacity. Since only the pruned models are transmitted between clients and the server, the communication cost is substantially reduced. For example, the pruning ratios applied to different client levels (Table 4) demonstrate how ATTENDING enables efficient model transmission.
>
> > Weakness 2: The disadvantages of using matrices and the advantages of ATTENDING are not well analyzed and compared theoretically and experimentally.
>
> **Response:** The disadvantages of binary mask matrices are summarized as follows:
> a) Binary mask matrices must be transmitted alongside model parameters, leading to additional communication costs, especially for large models and frequent FL rounds.
> b) In typical deep learning frameworks, such as PyTorch, computations involving binary mask matrices are inefficient because the pruned parameters are still considered during backpropagation (e.g., set to zero), resulting in unnecessary gradient calculations.
> c) Binary mask matrices make it difficult to align and aggregate pruned models from heterogeneous clients effectively, as demonstrated in prior works (e.g., Wang et al., 2020a; Jiang et al., 2022a).
>
> The advantages of ATTENDING are summarized as follows:
> a) By leveraging attention scores, ATTENDING directly determines which parameters to retain or remove, avoiding the need for mask matrices to track parameter states.
> b) The attention matching mechanism aligns pruned models based on their attention scores, mitigating the challenges of structural mismatch without relying on binary masks.
> c) Without binary mask matrices, ATTENDING reduces both communication and computational overhead, as only the necessary parameters are transmitted and updated.
> d) The empirical results presented in Tables 2 and 3 demonstrate that ATTENDING achieves superior accuracy while maintaining lower model footprints, indirectly highlighting the benefits of avoiding binary mask matrices.
>
> > Weakness 3 \& Question 2: The experiment part mainly focuses on accuracy, also lacks communication and storage results.
> > How many computation and communication resources do ATTENDING and baselines require?
>
> **Response:** Storage Resource Comparison: The proposed ATTENDING method significantly reduces the number of model parameters, as shown in Table 4, which directly translates to lower storage consumption.
> Computation Resource Comparison: While Table 4 shows FLOPs reductions, we acknowledge that this is a theoretical metric. To address this limitation, we evaluated the wall time elapsed during 100 FL rounds for ATTENDING and other methods. The results are presented below.
>
> | FL Algorithm  | FedAvg | FedDrop | FedProx | FedNova | Hermes | FedMP | FedGH | FedP3 | ATTENDING |
> |---------------|--------|---------|---------|---------|--------|-------|-------|-------|-----------|
> | Running Time  | 381s   | 342s    | 359s    | 173s    | 504s   | 555s  | 720s  | 532s  | 359s      |
>
> **Table**: Comparison of wall time elapsed during 100 FL rounds on the MNIST dataset.
>
> > Weakness 4: MNIST is used in two of the three Envs, but this dataset is too simple, and more complicated datasets are suggested to replace it.
>
> **Response:** The datasets used in our work (i.e., MNIST, CIFAR10, and CIFAR100) are widely used in the FL community as benchmarks to evaluate and compare algorithms. They provide a clear and standardized foundation for testing various methods under controlled conditions. Using these datasets allows us to focus on demonstrating the effectiveness of our proposed approach while ensuring comparability with prior work.

---

> ### Author Response · Authors · 2024-11-23
>
> > Weakness 5: Experiments should include more different alpha values regarding data heterogeneity and more different client configurations regarding system heterogeneity issues.
>
> **Response:** We recognize that varying alpha values could provide deeper insights into ATTENDING’s performance under different levels of data heterogeneity. We would like to highlight that, experiments on different alpha values have been provided in Appendix I.
>
> The client configurations used in our work are aligned with the convention settings in FL (Wang et al., 2020b; Mei et al., 2022; Alam et al., 2022). Using these configurations allows us to ensuring comparability with prior work.
>
> > Weakness 6: Figure 1 only compares ATTENDING with FedAvg; other baselines aiming for the data heterogeneity issue should also be included.
>
> **Response:** Figure 1 is designed to provide an intuitive and visual demonstration of how ATTENDING addresses the data heterogeneity issue by focusing on critical data features. We selected FedAvg as a baseline because it is one of the most widely used methods in FL and serves as a standard benchmark for comparison. Including FedAvg helps highlight the contrast in performance when no specialized techniques for addressing heterogeneity are applied. While Figure 1 focuses on the comparison with FedAvg, our experimental results in Sections 4.2 and 4.3 provide comprehensive numerical comparisons with other state-of-the-art methods designed to address data heterogeneity. These results demonstrate that ATTENDING achieves significant improvements in global model accuracy under non-IID settings, as shown in Tables 2 and 3.
>
> > Weakness 7: The limitations of ATTENDING should also be discussed in the paper.
>
> **Response:** The main limitations of ATTENDING are its current lack of support for various types of neural networks and multi-modality tasks. These limitations have already been discussed in Section 5, lines 483–485: ``Our future research endeavors will focus on exploring the implementation and extension of ATTENDING across various types of neural networks and multi-modality tasks, thereby enhancing its overall applicability."
>
> > Question 1: What is the intuition to calculate the threshold in the current way, i.e., equation (7)?
>
> **Response:** The threshold is designed to select the channels that have the least importance according to the attention mechanism. The attention scores reflect the importance of each channel in capturing relevant features from the data. By selecting the $\hat{\lvert E \rvert}$-th smallest attention score, we ensure that the least important channels (those with low attention scores) are pruned. This approach aims to retain the channels with the highest attention scores, which are deemed more crucial for model performance, and remove those that are less relevant.

---

### Official Review · Reviewer_poxA · 2024-11-03

**Soundness:** 2
**Presentation:** 3
**Contribution:** 1
**Rating:** 3
**Confidence:** 5

**Summary:**

The paper presents ATTENDING (personalized ATTENtive pruning enabled federateD learnING), an approach to tackle the dual challenges of system and data heterogeneity in Federated Learning (FL). The authors propose an attention module that leverages spatial and channel attention to improve learning across diverse data distributions. They also introduce an attentive pruning method that produces personalized models based on attention scores, allowing broader client participation. Finally, a specialized heterogeneous aggregation algorithm, combined with attention matching, enables efficient model aggregation. Together, the paper claims these innovations improve FL’s adaptability and performance on heterogeneous systems and datasets.

**Strengths:**

+The paper is well represented, and easy to read.

+The paper tries to address multiple challenges in FL: non-iid, heterogeneous client, resource constraints. All of them are interesting research directions in FL.

**Weaknesses:**

- The paper mixes of three techniques: attention, personalized pruning, heterogeneous aggregation. All proposed methods are largely align with prior work. The paper fails to demonstrate the novelty, not even any incremental improvements.

- The paper fails to show the significance of integrating the three techniques together. Why does the paper put the three techniques together? How can they interplay? What are the challenges of integration?

- The paper lacks theoratical analysis.

-  The evalution is weak. Please see the specific questions.

-  Related work is not sufficient. There are a lot of work in pruning, heterogieous devices for FL. The proposed method shall be compared with them.

**Questions:**

The paper is only compared with the methods for non-iid and model compression. How about the works in literature on heterogieous devices for FL and personazlied FL? Given that the proposed approach integrates three techniques, it would be more appropriate to compare it against existing works that also combine multiple techniques.

How are experiment settings determined? How are the model, dataset, pruning ratios,  α selected?

How many clients are used for MNIST dataset in talbe 2 and table 3? How is the average performance cacluated?

Why MNIST dataset is spllitted for 100 cleints, while cifar10 and cifar100 are splited for 10 clients?

“FLOPs” can not be used to determine computation consumption. It is just a theoretial justification when sparsification is used. Please run the algorithm on-device and show the actual running time. Because the method is complex, the actual computation time on real device may be longer than other approaches.

There is no result showing heterogeous devices. Since the method is propopsed for heterogeous devices. Please show such results on real device.

The paper only shows results on image data. Please justify the applications of image data on heterogeous devices for FL. Or the paper shall include results with diversified datasets.

The datasets used in evaluation are too simple, including MNIST, cifar10, and cifar100. Plus, they are not the dataset for heterogeneous FL, because the datasets have to be splitted artifically for the clients and non-iid.

Minor issue: please carefully select the Primary Area when submitting the paper, instead of just using "other topics in machine learning (i.e., none of the above)". There must be a more appropriate Primary Area for the paper.

---

> ### Author Response · Authors · 2024-11-23
>
> We sincerely thank you for your constructive and helpful comments. Below we address your concerns in order. The discussion here will be properly incorporated into a new version of our paper.
>
> > **Weakness 1:** The paper mixes of three techniques: attention, personalized pruning, heterogeneous aggregation. All proposed methods are largely align with prior work. The paper fails to demonstrate the novelty, not even any incremental improvements.
>
> **Response:** We would like to emphasize that our proposed framework is **NOT** a combination of existing approaches. To the best of the authors' knowledge, there has been **NO** prior work that proposes attention-guided federated model pruning methods or attention-guided heterogeneous model aggregation. We are open to discussing and comparing our work with studies that genuinely "largely align with" ours, should the reviewer provide specific examples. In the following, we address the concern regarding the novelty of our work:
>
> Novelty of the Attention Module:
> While attention mechanisms have been extensively studied in centralized deep learning, their application in FL remains underexplored, especially in resource-constrained edge environments. In our work, we designed a novel attention module tailored for FL. This module not only enhances feature extraction on non-IID data but also directly influences the personalized pruning and aggregation steps, which differentiates it from prior work where attention mechanisms were not integrated with such objectives.
>
> Personalized Attentive Pruning:
> The proposed pruning algorithm goes beyond traditional model pruning techniques by leveraging attention scores to guide the pruning process. This attention-driven approach ensures that the pruned models are highly personalized to the data and system capabilities of each client, unlike existing methods that rely on binary masks or uniform pruning criteria. This innovation enables low-end devices to participate in FL without compromising performance.
>
> Heterogeneous Aggregation with Attention Matching:
> We introduce a novel aggregation mechanism that uses attention matching to address the permutation invariance and structural differences between pruned models. This aggregation method mitigates the adverse effects of direct averaging in heterogeneous FL, which prior works have shown to degrade performance. The attention matching mechanism is simple yet effective in maintaining consistency among pruned models during aggregation.
>
> Demonstration of Incremental Improvements:
> Our experimental results (Tables 2 and 3) clearly demonstrate the superiority of our approach over state-of-the-art methods, achieving up to 11.3% better accuracy on non-IID data and reducing model footprints by 32%. This balance of efficiency and performance enhancement underscores the practical significance of our contributions.

---

> ### Author Response · Authors · 2024-11-23
>
> > Weakness 2: The paper fails to show the significance of integrating the three techniques together. Why does the paper put the three techniques together? How can they interplay? What are the challenges of integration?
>
> **Response:** We would like to highlight that this paper is **NOT** simply integrating existing techniques together. Please refer to the Response for Weakness 1 for further information.
>
> Rationale of the Three Key Components:
> The motivation behind attention mechanism, personalized pruning, and heterogeneous aggregation lies in their complementary roles in addressing the dual challenges of system and data heterogeneity in FL: The attention module is designed to capture non-IID data features and compute importance scores for model parameters. This lays the foundation for the personalized pruning step by providing data-driven insights into parameter relevance. Personalized pruning, guided by attention scores, ensures that each client operates on a model tailored to its data and system capabilities, enabling resource-constrained clients to participate effectively in FL. Heterogeneous aggregation integrates these personalized models back into a coherent global model using the attention matching mechanism. This step ensures that the aggregated model respects the structural differences introduced by pruning while minimizing accuracy loss.
>
> Interplay of the Components:
> The three components are inherently interdependent: The attention module generates importance scores that directly guide the pruning process, influencing the structure of personalized models. The personalized pruning step, in turn, produces heterogeneous local models with different architectures and parameter distributions, necessitating a specialized aggregation mechanism. The heterogeneous aggregation method uses attention scores to match and align pruned model parameters, effectively bridging the gap between diverse local models.
>
> Challenges of Developing These Components:
> Developing these components posed several challenges, which we addressed as follows.
> Balancing Model Performance and Resource Constraints: Designing the attention-driven pruning mechanism required careful calibration to maintain model accuracy while reducing computational and storage demands. Aggregation of Heterogeneous Models: The structural differences introduced by pruning made traditional aggregation techniques ineffective. We tackled this by proposing the attention matching mechanism to align model structures and mitigate permutation invariance issues.
>
> Significance of the Proposed Method:
> The experimental results (Sections 4.2–4.4) demonstrate the effectiveness of the proposed framework. For instance, the attention module significantly improves feature extraction, which enhances the performance of the pruning and aggregation steps. This design leads to state-of-the-art accuracy and resource efficiency, as shown in Tables 2–4.
>
> > Weakness 3: The paper lacks theoratical analysis.
>
> **Response:** It is important to note that extensive prior work has established the convergence of model pruning-based FL frameworks. Our proposed framework does not violate the convergence properties of FL. The comprehensive experimental results presented in our paper further support the convergence of our framework, demonstrating consistent performance improvements and stability across various experimental setups.
>
> > Weakness 4: Related work is not sufficient. There are a lot of work in pruning, heterogieous devices for FL. The proposed method shall be compared with them.
>
> **Response:** To the best of our knowledge, we have conducted a systematic literature review, incorporating **57 references** for a discussion of related work.
> We are open to including more if the reviewer can specify what we missed.
>
> > Question 1: The paper is only compared with the methods for non-iid and model compression. How about the works in literature on heterogieous devices for FL and personazlied FL? It would be more appropriate to compare it against existing works that also combine multiple techniques.
>
> **Response:** We would like to highlight that this paper is **NOT** simply integrating existing techniques together. Please refer to the Response for Weakness 1.
> In the evaluation section, we have included 8 fresh SOTA methods for comparison. We believe these comparisons demonstrate the advantages of the proposed method.
>
> > Question 2: How are experiment settings determined? How are the model, dataset, pruning ratios, $\alpha$ selected?
>
> **Response:** The determination of experiment settings has been provided in Section 4, lines 330-367 in detail, including the selection of the model, dataset, pruning ratios, and $\alpha$.
>
> > Question 3: How many clients are used for MNIST dataset in talbe 2 and table 3? How is the average performance cacluated?
>
> **Response:** The client number and average performance calculation method have been described in Table 1 and lines 346-347 in detail.

---

> ### Author Response · Authors · 2024-11-23
>
> > Question 4: Why MNIST dataset is spllitted for 100 cleints, while cifar10 and cifar100 are splited for 10 clients?
>
> **Response:** The client numbers are aligned with the conventional setting in FL studies. The rationale behind these settings is described as follows:
> The MNIST dataset, being relatively simple and smaller in size, allows for experiments with a larger number of clients. Splitting MNIST among 100 clients provides a more granular simulation of FL in large-scale scenarios, where many edge devices with limited data participate in the FL process. This setting also helps evaluate the scalability and effectiveness of our proposed methods in scenarios with higher client heterogeneity.
> CIFAR10 and CIFAR100 are significantly more complex datasets compared to MNIST, containing higher-dimensional images and more classes. In real-world FL settings, training on such complex data is often distributed among fewer, more capable clients, as the computational and storage requirements are higher. Splitting these datasets among 10 clients reflects such a scenario, where each client possesses a relatively larger portion of the dataset, mimicking environments with moderate client heterogeneity and richer data per client.
>
> > Question 5: “FLOPs” can not be used to determine computation consumption. Please run the algorithm on-device and show the actual running time.
>
> **Response:** FLOPs were used as a standardized metric to compare the computational demand of different methods under the same conditions. However, we agree that actual running time measurements are more reflective of practical performance.
> Benefiting from the substantial reduction in model footprints, the actual computation time of ATTENDING is not expected to exceed that of other approaches. For instance, using the MNIST dataset as an example, we evaluated the wall time elapsed during 100 FL rounds for ATTENDING and other methods. The results are presented below.
>
> | FL Algorithm  | FedAvg | FedDrop | FedProx | FedNova | Hermes | FedMP | FedGH | FedP3 | ATTENDING |
> |---------------|--------|---------|---------|---------|--------|-------|-------|-------|-----------|
> | Running Time  | 381s   | 342s    | 359s    | 173s    | 504s   | 555s  | 720s  | 532s  | 359s      |
>
> **Table**: Comparison of wall time elapsed during 100 FL rounds on the MNIST dataset.
>
> > Question 6: There is no result showing heterogeous devices. Please show such results on real device.
>
> **Response:** The results on heterogeous devices have been provided in Table 2, Table 3, Figure 4, and Figure 6. The detailed heterogeous client configuration is also provided in lines 348-353.
>
> > Question 7: The paper only shows results on image data. Please justify the applications of image data on heterogeous devices for FL. Or the paper shall include results with diversified datasets.
>
> **Response:** Image data is commonly used in FL research due to its relevance in many real-world applications involving heterogeneous devices. Such as smartphone applications, healthcare, and smart IoT devices. By focusing on image data, our study directly addresses practical scenarios where FL on heterogeneous devices is highly applicable.
>
> > Question 8: The datasets used in evaluation are too simple. Plus, they are not the dataset for heterogeneous FL, because the datasets have to be splitted artifically for the clients and non-iid.
>
> **Response:** These datasets are widely used in the FL community as benchmarks to evaluate and compare algorithms. They provide a clear and standardized foundation for testing various methods under controlled conditions. Using these datasets allows us to focus on demonstrating the effectiveness of our proposed approach while ensuring comparability with prior work.
> The non-IID data partition approach is a common practice in FL research, as real-world datasets with naturally non-IID distributions are not always readily available. Specifically, we adopted the Latent Dirichlet Allocation (LDA) strategy, a widely used method for simulating realistic heterogeneity by controlling the level of non-IIDness.
>
> > Question 9: Minor issue: please carefully select the Primary Area when submitting the paper. There must be a more appropriate Primary Area for the paper.
>
> **Response:** We thank the reviewer for pointing out the importance of carefully selecting the Primary Area during submission.
> We will ensure to select a more fitting Primary Area to better align with the focus of our work.

---

### Comment · Area_Chair_W8Zo · 2024-12-01

Dear Reviewers,

This is a reminder that December 2 is the final day to post feedback to the authors.

Your input is critical to the review process. If you have not yet responded, we kindly encourage you to confirm whether you have reviewed the rebuttals. Additionally, if you have any remaining concerns or clarifications, please share them with the authors before the deadline.

Thank you for your valuable contributions.

Best,

Your AC

---

### Note · Authors · 2025-07-14

I have read and agree with the venue's withdrawal policy on behalf of myself and my co-authors.

---

### Meta-Review · Area_Chair_W8Zo · 2024-12-22

**Metareview:**

This paper proposes ATTENDING, an FL approach that addresses both data and system heterogeneoties. The paper demonstrates improvements in accuracy and reductions in model footprint compared to existing baselines. Its major weaknesses lie in the technical contributions. The work is inadiquent to differential the proposed method from prior work. Additionally, the claim to address system heterogeneity is weakened by the requirement for all clients to initially run the original model, which contradicts the goal of resource-constrained environments. Privacy concerns and the potential misalignment caused by attention-matching during aggregation were also noted but insufficiently addressed. The experimental evaluation would benefit from incorporating more complex datasets and realistic FL settings to provide stronger support for the method's contributions.

**Additional Comments On Reviewer Discussion:**

The reviewers’ concerns focused on the technical contributions and novelty, the experimental setup's ability to substantiate the heterogeneity claim at the system level, privacy implications, and the lack of theoretical analysis. While the authors addressed some of these points during the rebuttal, privacy implications remained a critical unresolved weakness, and the experimental setup did not convincingly address concerns about realistic FL scenarios. Given these unresolved issues, particularly around novelty, system heterogeneity, privacy, and experimental rigor, the paper does not meet ICLR's high standards and is not recommended for acceptance.

---

### Decision · Program_Chairs · 2025-01-22

Reject